# Relevance of Sociocultural Inequalities and Parents’ Origins in Relation to the Oral Health of Preschoolers in Lanzarote, Spain

**DOI:** 10.3390/healthcare11162344

**Published:** 2023-08-20

**Authors:** Beatriz Prieto-Regueiro, Gladys Gómez-Santos, Daniele Grini, Laura Burgueño-Torres, Montserrat Diéguez-Pérez

**Affiliations:** 1Primary Care Odontoestomatologist, Health Services Management of Lanzarote’s Health Area, 35500 Arrecife, Spain; bprireg@gobiernodecanarias.org; 2Health Promotion Service, General Directorate of Public Health of the Canary Islands Health Service, 38005 Santa Cruz de Tenerife, Spain; ggomsan@gobiernodecanarias.org; 3Department of Preclinical Dentistry, Faculty of Biomedical and Health Sciences, Universidad Europea de Madrid, Villaviciosa de Odón, 28670 Madrid, Spain; daniele.grini@universidadeuropea.es; 4Dental Clinical Specialties Department, Faculty of Dentistry, Complutense University of Madrid, 28040 Madrid, Spain; lbtorres@ucm.es

**Keywords:** inequality, socioeconomic, sociocultural, origin, educational level, oral health, dental caries, pre-school, children

## Abstract

Approaching inequalities to achieve health equity requires joint action. Early childhood caries affects disadvantaged population groups. The objective of this study was to determine the relevance of sociocultural inequalities and parental origin with respect to oral health in preschool children in Lanzarote. A transversal epidemiological study was carried out. Sociocultural data and information about parental origin were collected via a questionnaire. The decayed and filled teeth (dft), decayed teeth (dt), filled teeth (ft), restorative index (RI), plaque index (PI) and gingival index (GI) were obtained via an investigator’s examination. Statistical analysis of the data indicated that children of semi-skilled manual workers (28.15%) had the highest prevalence of caries (46.9%). Additionally 43.7 of the mothers had second grade and first cycle studies. When relating the medium and high level of education, there were statistically significant differences in relation to the cod index (*p* = 0.046). When the origin was foreign (48.4%), preschoolers presented 10.7% more active or untreated caries than Spaniards (*p* = 0.038). Low socioeconomic cultural level and foreign origin is associated with a more deficient state of oral health in preschoolers. Oral health programs are needed to minimize social inequalities.

## 1. Introduction 

Achieving equity in health is a social justice objective that contributes to the development of more prosperous societies. Social inequalities in health are present in all countries, in all areas and throughout the population. Dealing with them requires the joint action of different sectors; in addition to the health sector, the participation of the population is required. It is also essential to pay special attention to actions related to the context and social determinants of health to ensure health equity [1]. A social disadvantage at any period of life can have a lasting detrimental impact on one’s quality of life in middle age [2].

The World Health Organization (WHO) glossary of Health Promotion terms in 2021 defines health determinants as the range of personal, social, economic and environmental factors that determine the healthy life expectancy of individuals and populations [3].

The model developed by Dahlgren and Whitehead of the WHO European Regional Office shows the hierarchy of determinants in onion layers, which is one of the most widely known models [4]. Furthermore, in our country there is a conceptual framework adapted to the national setting by the Commission for Reducing Inequalities [5,6] in Health in Spain, based on the framework developed by the WHO Commission on Social Determinants [7], which presents two main hierarchies of determinants (Figure 1). The intermediate determinants, in a lower hierarchy, directly influence health and the National Health System (SNS) takes on the role of mediator or cushion of the consequences of the disease. The structural determinants would be in the higher hierarchy, influencing health through intermediaries. In first place is socioeconomic position, which is determined by education, occupation, earnings, gender and ethnicity. On top of this, another consideration is the socioeconomic and political context (governance, policies, culture and values of society), which configures, generates and maintains social hierarchies.

The unequal distribution of social determinants causes an unequal distribution of health. Social inequalities affect all territorial levels: by country, by region, within countries, by municipality and by neighborhood within the same locality [8]. 

In Spain, the term social inequalities in health is mostly used, although at the international level, the term inequities is used, being equivalent terms in terms of meaning. Studies on social inequalities in health have shown on multiple occasions that certain groups, such as the population with fewer economic resources, women and immigrants, have health indicators that denote a worse situation [9].

In this regard, several studies have been carried out in Spain that show clear inequalities in the use of health services by immigrants [10,11].

Regarding oral health, early childhood caries continues to be a pathology of worldwide relevance that mainly involves the most disadvantaged population groups; in this sense, there is no disagreement among researchers [12,13] although one must bear in mind the perspective of dynamism in terms of the development of our societies and their lifestyles.

The socioepidemiology of oral health issues, as well as indicators related to socioeconomic status, such as the social class to which children who suffer the consequences of carious lesions belong, together with the education level of their parents, their work situation, influenced by monetary income and the area in which their family lives, promote different levels of exposure and vulnerability with respect to carious pathology [12,13,14,15].

These social inequalities within a community and between different populations represent a great challenge for political leaders; therefore, overlooking this fact would be prejudicial to achieving optimal public health in child populations [16]. Early detection of early childhood lesions associated with these population groups would allow preventive and therapeutic measures for their control, avoiding an increase in their prevalence over the years, since it has been found that the highest frequency of these lesions is related to preschool growth [17,18,19]. 

In children, not only social advantage but also rurality and caregiver education are consistent predictors [20]. Educational level and parental origin [13,21,22,23,24] can also be risk elements for early childhood caries, which makes this updated study relevant. 

This research has been justified based on the fact that in the Spanish preschool population, the impact of all these factors on oral health has yet to be determined, thought it has been evaluated based on the presence of decayed and filled teeth and the state of oral hygiene evaluated in terms of plaque and gingival indices. Furthermore, with respect to this last aspect, there is little information in the scientific literature on the preschool population. Another aspect to be taken into consideration is that more detailed knowledge of the characteristics of the most disadvantaged population group would make it possible to redirect part of the economic investment to improve the health of preschoolers. 

Therefore, the aim of this research was to determine the relevance of sociocultural inequalities and other factors, such as the origin of parents, in relation to the oral health of Lanzarote children between 3 and 5 years of age.

## 2. Materials and Methods

### 2.1. Population and Study Design

A transversal epidemiological study was carried out in preschoolers of both sexes aged 3 to 5 years. The population of the municipality of Arrecife, which in the primary care setting is distributed in two Basic Health Zones (ZBS), Arrecife I (ACEI) and Arrecife II (ACEII), was randomly selected. To this end, an envelope containing a name was extracted, which had been previously deposited in an urn that included all the municipalities of Lanzarote (each one of them in its corresponding envelope). In the absence of data on the prevalence of caries in preschoolers in this health area, the sample size was calculated based on the documented cases of this disease in the temporary dentition of the school population of Lanzarote, for an expected proportion of 70%, following the pattern of an upward trend.

The sample size calculation for the total population of 2018 persons, according to the health card database in May 2015, with an expected caries prevalence of 70%, confidence level of 95%, design effect 1, precision of 5%, and proportional distribution, resulted in 175 patients for the ZBS ACE I, and 108 patients for the ZBS ACE II. 

Patients aged 3 to 5 years with a Canary Islands Health Card who attended the pediatric service of the ZBS of Arrecife with a prior appointment were randomly recruited. The study included those whose parents/guardians accepted and signed informed consent; the study information and a personal data release form were included. Exclusion criteria included parental refusal to participate at any time during the study, and/or lack of collaboration of the minor. After these selection criteria were applied, of the 18 missed appointments, 5 were related to absence from the scheduled appointment and the remaining 13 to the lack of collaboration of the preschooler in the dental examination.

A validated clinical form was used for the collection of the variables, where the data on filiation and anamnesis, an oral health questionnaire and an oral examination form were recorded.

### 2.2. Questionnaire

Periodically, three days a week, preschoolers who attended the pediatrician’s office and satisfied the selection criteria were randomly selected. After informed consent and personal data consent were obtained, the affiliation data were recorded and a socioeconomic questionnaire was completed by the researcher in a structured interview with the parents or guardians. The estimated duration of the interview was 5 to 10 min.

### 2.3. Socio-Demographic Variables

In addition to analyzing sex and age, the socioeconomic level of the family was studied. In this sense, the social level was determined according to the occupation of the head of the family; that is, the person who contributed most regularly to the household budget. For this purpose, the abbreviated social class classification based on occupation proposed by the Spanish Society of Epidemiology working group was used [25].

The categories are as follows:Category I includes managers in public administration and companies with 10 or more employees and professions associated with second- and third-cycle university degrees;Category II includes managers of companies with fewer than 10 employees, professions associated with a first-cycle university degree, technicians, artists and athletes;Category III includes clerical employees and administrative and financial management support professionals, personal and security services workers, self-employed workers and supervisors of manual workers;Category IV is subdivided into IVa, which refers to qualified manual workers, and IVb, which refers to semi-qualified manual workers;Category V includes unqualified workers.

The maternal level of education was defined as a cultural indicator of the family, since mothers were present in almost all the families, minimizing the cases lost, and also because of their greater influence on children’s health. When the father was the only adult in the household, the paternal level of education was taken into account. To measure this variable, the highest level of education attained was used [25,26]:Category 1 refers to people with no studies;Category 2 refers to first-degree studies;Category 3 refers to second-degree studies, first cycle;Category 4 refers to second-degree studies, second cycle;Category 5 refers to third-degree studies, university.

To relate the level of education to caries indicators and oral hygiene indexes, low level of education was recoded to include categories 1 and 2, medium level of education to include categories 3 and 4, and high level of education to include category 5.

Due to the intensification of the immigration phenomenon in the Autonomous Community of the Canary Islands, it was decided to record the origin of the minor. To determine this last variable and with the purpose of achieving greater variability in the sample, due to the short age range studied, the Oral Health Survey of Preschoolers carried out in Spain in 2007 [27] was taken as a reference, considering the child to be of Spanish origin when both parents were Spanish and of foreign origin if the origin of one of them was from a country other than Spain. No origin was recorded if this information was unknown for at least one of the two parents. In relation to foreign nationality and in accordance with the criteria of the 2003 National Health Survey, the country of origin of the parents of the subjects studied was grouped into 7 categories [28]:Category 1: A country of the European Union;Category 2: Another European country;Category 3: Canada or the USA;Category 4: Another country in America;Category 5: A country in Asia;Category 6: A country in Africa;Category 7: A country in Oceania.

### 2.4. Clinical Exams and Oral Examination

Two days a week, 7 patients per day were seen in the Oral Health Unit (USO). The oral examination was carried out with the preschooler seated in the Fedesa JS 500 dental chair, with the neck extended and the examiner behind them in a standing position. The estimated examination time was 20 to 30 min per preschooler, and the instruments used consisted of a number 5 intraoral flat oral mirror and the WHO periodontal probe. The WHO recommended methodology was used for both the systematic oral examination and the diagnostic criteria for caries detection [29]. The subsequent indices were found:Index dft: Number obtained after recording all the erupted primary teeth that presented carious lesions, as well as those that were filled;Index dt: Number obtained after recording all the erupted primary teeth that only presented caries lesions;Index ft: Number obtained after recording all primary teeth that only presented fillings;Index dfs: Number obtained by the sum of all the coronal dental surfaces of the deciduous teeth presented in the oral cavity, both with caries lesions and with fillings, divided by the total number of explored patients;Index IR: Percentage resulting from the division between the value obtained by the index ft and dft multiplied by 100.

To determine the dental plaque of each tooth, the simplified Silness and Löe Plaque Index (PI) [30] was used, starting with the distal surface of the last molar, following the sequential vestibular, mesial and lingual or palatal path. To detect the plaque that was not visible to the naked eye, the IPC probe, which ends in a sphere, was used to sequentially follow the contour of the gingival third. The index teeth studied were the permanent upper right central incisor or deciduous upper right central incisor, deciduous upper right canine, deciduous lower right first molar and the last four molars present in each quadrant. The values obtained were as follows:0: absence of plaque in the gingival area;1: plaque not visible, but can be removed from the gingival third of the tooth with the aid of a probe;2: moderate plaque accumulation in the gingival area visible to the naked eye;3: abundant plaque in the gingival area and even covering the adjacent tooth.

To measure gingival health, the simplified Löe and Silness gingival index (GI) [30] was used by introducing the rounded tip of the WHO probe into the gingival sulcus of the distal aspect of the last molar, following a continuous movement towards the vestibular, mesial and lingual or palatal areas of the index teeth mentioned in the previous section. The values used to determine the different degrees of gingival health were:0: no inflammation: normal gingiva;1: mild inflammation with slight change in color and little change in texture, without bleeding on probing;2: moderate inflammation with redness and shiny appearance, edema and moderate hypertrophy, bleeding on probing;3: severe inflammation with marked redness, edema and pronounced hypertrophy, spontaneous bleeding and ulceration.

### 2.5. Calibration

The clinical examination was performed by a single examiner, who had been previously trained and calibrated. Intraobserver variability control was performed with duplicate examinations, re-examining 10% of the sample, in a period of less than 1 week between the first and second examination. The examiner was evaluated with a second measurement for the oral examination variables through the kappa statistic, obtaining very high values all above 0.9 and a large number with kappa = 1, which indicated a very high degree of intraexaminer agreement.

### 2.6. Etical Aspects

This study was approved by the Health Sciences Research Committee of the Universidad Europea de Madrid and by the Clinical Research Ethics Committee of the Complejo Hospitalario Universitario Insular-Materno Infantil (CEIC-CHUIMI) and complies with the ethical precepts established in the Declaration of Helsinki as well as the Law on Patient Autonomy (41/2002) and the Organic Law on Data Protection (15/1999).

### 2.7. Statistical Analysis

The data, which were collected in a Microsoft Excel spreadsheet, were analyzed using SPSS 25.0 for Windows. For the descriptive analysis of the quantitative variables (caries index), the mean, the standard deviation, and the 95% confidence interval were used.

Meanwhile, for the qualitative variables (socioeconomic, cultural variables, origin of the parents, and plaque and gingival index), the frequencies and percentages of the categories were used. In addition, the Chi-square test was used to compare the independence or influence between two qualitative variables, the ANOVA analysis of variance for the comparison of multiple groups, and the Bonferroni test for multiple comparisons between pairs of groups. A value of *p* < 0.05 was considered statistically significant.

## 3. Results

### 3.1. Sociodemographic Characteristics of the Sample

The total number of children studied after the selection criteria was applied was 343 preschoolers, distributed by sex into 175 boys and 168 girls. In total, 99 of the preschoolers were 3 years old, while 122 were within the 4- and 5-year-old age group. The mean age of the study population was 3.27 ± 0.28, 4.39 ± 0.30 and 5.39 ± 0.33, respectively. The sociodemographic characteristics of the parents of the studied preschoolers, distributed by sex and age, are shown in Table 1.

In terms of social status, semi-skilled manual workers were the most common group (28.15%) in the total sample; while managers in public administration and companies with 10 or more employees and professions associated with second- and third-cycle university degrees were the least common (5.2%). In relation to educational level, only 0.6% of the parents were illiterate and the highest percentage (43.7%) of the mothers had second-cycle and first-cycle degrees. Foreign origin corresponded to 48.4% of the studied population (Figure 2).

Taking into account the sex and age of preschoolers, a higher percentage of 3-year-old Spanish boys belonged to the most prevalent group in terms of social status. And a higher percentage of 5-year-old Spanish girls belonged to the most frequent group in relation to educational level.

### 3.2. Oral Health Indicators Associated with Sociodemographic Characteristics

The results obtained with respect to the number of decayed and filled teeth by sex and age are shown in Table 2 and Figure 3. The dft and dt indexes were statistically significant when related to preschool age (*p* < 0.001). Such significance was not observable with respect to sex.

The highest percentage of preschoolers with decayed and filled teeth were 5-year-old boys.

When the dft and dt indices were related to family social level (Table 3), semi-qualified manual workers were the ones with the highest prevalence of caries in their children (46.9%) compared to social class I, II (20.0%) (*p* = 0.016). This result was due to the carious component (45.8% vs. 18.0), (*p* = 0.007).

When considering the social level, statistical significance was obtained for the mean of the dft index (*p* = 0.006) and the mean of the carious component (*p* = 0.006). The Bonferroni multiple comparisons test showed how these statistically significant differences were present between parents of social level I, II and those of level IVb.

When relating educational level and dft index (Table 3), the Bonferroni multiple comparisons test showed significance when comparing the medium education level (1.72) with the high education level (0.75), (*p* = 0.046).

Preschoolers with foreign parents had 10.7% more active or untreated caries than those of Spanish origin (*p* = 0.038) (Table 3 and Figure 4).

The percentage of decayed teeth was always higher that of than filled teeth, regardless of the social status, educational level and origin of the parents.

The restorative index was related to social status (Table 4), educational level and origin of the parent (Table 5 and Table 6). The lowest restoration index was recorded in category IVb (6.15%) and the highest was associated with parents who were unqualified workers (19.05%).

The low educational level of the parents corresponded with a higher rate of restoration (25%). Preschoolers with parents of Spanish origin had a restorative index of 13.75%.

In none of the cases were statistically significant differences observed between the restoration index and the social status, educational level and origin of the parents.

On studying the PI and GI relating them to the sociodemographic characteristics of the parents (Table 7), the results indicated that the grade 1 PI was the most frequent in the study population and was related to a higher percentage of parents who were semi-qualified manual workers (87.5%). This was also observed when studying the grade 1 GI. In relation to educational level, the children of parents with a medium level of education presented the highest percentage of grade 1 PI (91.8%) and grade 1 GI (87.2%). Again, Spanish origin represented the highest percentage in both indexes (89.3% vs. 85.3%).

## 4. Discussion

Early childhood caries is a global public health problem that also impacts the quality of life of both the child and their family, and in more severe cases, it can affect school performance. Although it is to be expected that the most disadvantaged populations would bear a greater burden in this respect, the difficulty involved in being able to cover all the most disadvantaged population groups, taking into account the schooling of the parents and their origin, encourages us to investigate this aspect in the Spanish population in order to establish a system of health policies that specifically emphasizes a group with greater dental needs, which would also allow other countries to have greater control of the disease at a global level. This includes strategies to reduce social inequalities in oral health programs through the integration of educational programs in schools, which facilitate toothbrushing with fluoride toothpaste, educating families to promote oral hygiene at home [31] and controlling the social determinants from the preventive programs that are developed in primary care [32].

This study highlights the fact that parents who are semi-skilled manual workers represent the highest percentage (28.1%), as well as do mothers with a second or first degree (43.7%). However, the population frequency by origin is very similar in the selected sample, as 51.6% are Spanish. The high percentage of parents of foreign origin (47.2%) reflects the high level of migration in the population of Lanzarote, partly as a result of the arrival, by sea or air, of people without Spanish residence in search of a better life.

As expected, and in agreement with other studies [17,18,19,20,33,34,35], the mean of the dft, dt and ft indexes studied increased with age (dft = 0.91;1.14; 2.38 and dt = 0.88; 0.99; 2.25), and this difference was statistically significant (<0. 001), since the risk of a child having a greater number of caries lesions, if not prevented with preventive measures, will obviously increase over the years, since caries lesions are part of a continuous and cumulative pathological process. However, as in the work of other researchers, no significance was found in relation to gender when studying these indexes [18].

In general, and when these indexes are related to the socio-demographic characteristics of the parents, our results show that children of a low social level (IVb) have significantly higher caries indicators than those of a high social level (I, II). There is a clear social inequality in the oral health of preschool children. In the 2007, 2010 and 2015 national surveys, the unfavorable effect of low social status was observed for all caries indicators [27,36,37]. Inequalities related to social status were also found in the three epidemiological studies carried out in schoolchildren in the Canary Islands at the level Autonomous Community. [38].

In studies in developed countries, high values of carious surfaces, missing teeth due to caries and fillings were observed, as well as a higher prevalence of caries in children whose parents had a lower occupational range of work or lower income, indicative of a lower social level [17,18,19,39,40,41,42,43,44,45,46,47], which is why some studies confirm the significant predictive role that the economic factor has on dental caries [12].

According to Folayan et al., preschool children whose mothers have higher incomes use dental services more frequently [48].

Recent studies reflect how all social classes have a similar potential to develop caries, and although socioeconomic class has been considered a determinant factor, it is families with a poorer socioeconomic position that have a higher susceptibility to more severe stages of disease [17]. However, some authors have found weak evidence of the association of socioeconomic behavior factors and oral health [49].

When analyzing the results obtained in relation to the dft, dt and ft indices and their association with the mothers’ education level, it is interesting to note that all the indices were higher in preschoolers whose mothers had a medium level of education, compared to those with a lower level of education whose children presented lower indices. These results differ from those of other studies in which children whose parents had a secondary or primary level of education presented more unfavorable values with respect to caries indicators [12,20]. According to other researchers, a high level of education of mothers is a protective factor against dental caries [21,34]. Since, in the population of this study the lower educational level of the mother does not affect the greater presence of caries and fillings in the primary dentition, it is believed that these mothers may spend more time with their children and may be more aware of the follow-up of activities related to adequate oral health education. On the other hand, Yazdani et al. found that the parents’ university education (*p* < 0.05) was a protective factor against dental caries [50].

In economically advantaged countries and in contrast to the results, a statistically significant social gradient has been found among parents with a low level of education, both for a higher prevalence of caries [19,35,39,40] and for a higher dft [51] and dmfs (number of decayed and filled surfaces) index [42,44], with low maternal school education being a determining factor [52]. Some systematic reviews [13] reflect the strong association between oral health status and the socioeconomic position of individuals in populations in developed countries, and also indicate that the difference is more relevant when the level of education with our results is low, a fact which does not fit our results.

According to some researchers, the sociocultural factor related to the caries rate depends on the rural or urban environment, since the social inequalities associated with low family income with respect to caries were greater in preschoolers living in urban areas and school inequality resulting from the low level of education of the mothers was greater in rural areas [14].

Current research, in contrast to ours, highlights maternal education as the strongest predictor of dental caries, more so than the less relevant socioeconomic factor [53]. In this regard, in agreement with Marquiller et al., the most relevant factor is the level of parental knowledge about oral health, which can be significantly associated with caries indexes [15]. In Lanzarote, it was observed that the preschool children of foreign parents had 10.7% more active caries than those of Spanish origin (*p* = 0.038), and this difference between the two origins was also significant for the dft and dt index (*p* = 0.003 and *p* = 0.001, respectively). However, it is noteworthy that preschool children of foreign origin had higher means for all the indexes studied (2.02 vs. 1.08; 1.97 vs. 0.92), with the exception of the ft index, which was lower (0.05 vs. 0.16). This may be due to the fact that a foreign parent does not seek dental care as often as a Spanish parent.

As supported by the results obtained in this investigation, studies in countries with frequent immigration show a statistically significant influence of this factor on caries indicators. Thus, Steckesén-Blicks et al. [22,54] report that the prevalence of caries and the dmfs index were higher in the foreign population than in the Swedish natives; Skiee et al. [55] observed, as we did, that the dft and dmfs indexes and their carious components (carious tooth and carious surface, respectively) were higher in foreigners, with the obturated component being higher in the natives. Ferro et al. [23,56], Wigen et al. [24,57] as well as Baggio et al. [46] also reported a higher prevalence of caries in immigrants. In relation to this fact, it is believed that our foreign population may have different eating habits to the Spanish population and this may be a determining factor in making this difference evident.

Additionally, in the national surveys of 2007, 2010 and 2015, a clear and significant influence of the origin of the parent on the prevalence of caries and active caries and on the dft/dmft index was determined, with higher percentages and means in preschoolers of foreign origin [27,36,37,58]. According to Nota et al. [59] the foreign nationality of the mother, in this case non-Italian (*p* = 0.001) and a family income of less than a certain amount of euros per year (*p* = 0.002) significantly affect the caries experience of the preschooler.

The scientific literature does not reflect the association between the restorative index in preschoolers and their sociodemographic characteristics. In this study, the highest percentage of this index (19.0%) was observed among preschool children whose parents were unqualified workers, with a low level of education (25.0%) and of Spanish origin (13.7%).

Very few studies [35,60] have focused on the PI in the preschool population; in fact, the association of this index with either socioeconomic or educational factors in children of this age group has not been found in the scientific literature. Elamin et al. [60] studied the PI associated with parental origin in Emirati children aged 18 months to 4 years. In their investigation, this index was found to be much higher in Emirati children (1.8 ± 1.0 vs. 0.9 ± 1.0 in non-Emirati children), this difference being statistically significant. In general, the values obtained in this study were higher in Spanish children than in foreign children, but this difference was not significant and it is not known why this difference may occur.

No evidence was found in the scientific literature regarding the GI and its relationship with sociodemographic characteristics in the preschool population. The highest percentage of this index was observed among semi-qualified manual parents, average mother’s education level and foreign origin.

The Spanish population lives, on average, for longer and in better health than previous generations thanks to advances in the living conditions of the population and better access to resources and services. However, the study of inequalities in health has shown how socioeconomic position, gender, ethnicity or territory are factors of inequality with a great impact on the health of the population [32].

Reducing inequalities in child health is a key area to enhance health equity. There is widespread evidence of the importance of the early years of life in reducing social inequalities in health; inequalities in the childhood development stage will lead to inequalities in adult life and, consequently, to the creation of a cycle of intergenerational disadvantage. Today, studies reveal that many of the challenges faced by the adult population, such as mental health problems, obesity or heart disease, have their roots in early childhood. Interventions in the early years provide health benefits that last a lifetime, and societies that invest in children and their families in the early years of life have higher educational attainment, better health status, and lower health inequalities [5].

Prevention is one of the main interventions to be managed in primary care, and it is carried out through various programs based on scientific evidence. These programs should be evaluated throughout the whole process to identify possible inequities in their development. Tanahashi’s effective coverage model (Figure 5) [32] is useful to identify, at each key stage of the care process, why some groups access and benefit from health services and others do not. In this way, it is possible to identify barriers that impede the target population, or a segment of it, from making appropriate use of the program or health service offered and the facilitating resources, which are factors that help the target population to make appropriate use of the program, including those that make it possible to overcome the barriers to accessing health services and achieve effective use. These barriers and facilitators can occur at each stage of health service provision [1,8].

Currently, the portfolio of services of the Oral Health Program of the Canary Islands for preschoolers is very limited, and includes only dental check-up and extraction, so that only those families who can afford to pay for it can receive preventive and conservative treatments in private clinics [61]. In its future update, it is necessary to include tools that help to control inequalities, including the appropriate indicators to evaluate whether inequities are decreasing over time.

The methodological guide for integrating Equity in Health Strategies, Programs and Activities prepared by the Ministry of Health [32] offers instruments and support for preventive programs developed in Primary Care to minimize social inequalities.

The strength of this research lies in the fact that there are not many studies in the Spanish preschool population that attempt to relate socio-cultural aspects to parental origin. Another strength is that it studies the association between the restorative index and the gingival index with sociodemographic factors.

In terms of this study’s limitations, it should be highlighted that it does not reflect teeth lost due to caries, as the real reason for the loss was unknown, nor did it take into account the presence of caries. Neither was the presence of non-cavitated caries or white spot caries considered, the inclusion of which could increase the mean of the caries indicator indexes and perhaps alter the conclusions of our study.

## 5. Conclusions

The values of untreated teeth and the dft index are significantly higher in preschoolers of a low social level (IVb) compared to those of a high social class (I, II). However, the high restorative index is much more prevalent in very low social classes (V). Regarding oral hygiene in preschoolers, the lowest social level (V) has the highest percentage of PI grade 3. Also, a low social level (IVb) is associated with a higher prevalence of gingival index grade 2.

The mean educational level of the mother of a preschooler is significantly related to the mean of the dft and dt indexes. However, the highest percentage of restored teeth was observed in preschoolers whose mother had a very low level of education. The highest percentage of PI and GI of grade 2 is related to a medium level of education. The foreign origin of the parents is significantly associated with a higher prevalence of the dft index. There are more Spanish children with obturated teeth and their oral hygiene is poorer.

The results obtained in this study indicate the significance of developing and implementing an oral health protocol and/or program in the primary care setting that minimizes the inequalities observed due to their origin and social level. To achieve this, it is necessary that all professionals who participate in oral health programs recognize and internalize the problem of inequality in health care, identifying in their patients the situations and personal and social characteristics that can condition it. To achieve this, the training of professionals and the availability of resources and instruments are essential to monitor the inequalities generated by social determinants, with indicators that measure their evolution over time.

Likewise, it is necessary to advance as a priority in the implementation of comprehensive actions and policies that ensure good oral health from early childhood from the different levels of government, promoting equal development opportunities for this population.

## Figures and Tables

**Figure 1 healthcare-11-02344-f001:**
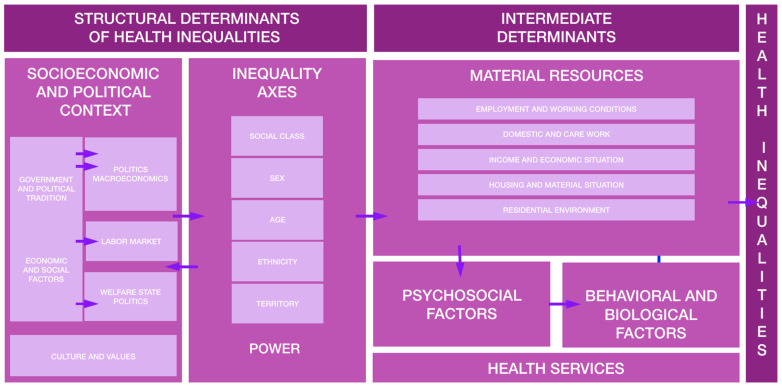
Conceptual framework of the determinants of social inequalities in health: Commission to reduce health inequalities in Spain (2010). Adapted from Soler and Irwin [5].

**Figure 2 healthcare-11-02344-f002:**
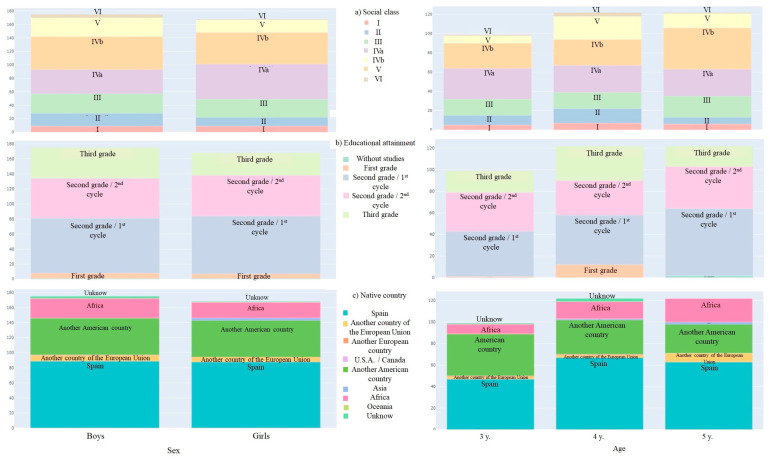
Graphical representation of social status, educational level and country of origin of parents by sex and preschool age range.

**Figure 3 healthcare-11-02344-f003:**
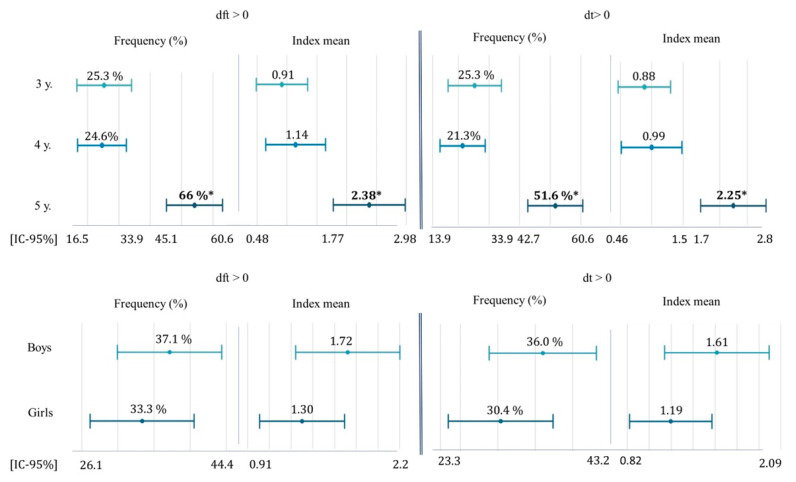
Graphical representation of the frequency, mean and 95% confidence interval of the dft index and its carious component (dt) in the total sample by age range and sex (*p* < 0.05 *).

**Figure 4 healthcare-11-02344-f004:**
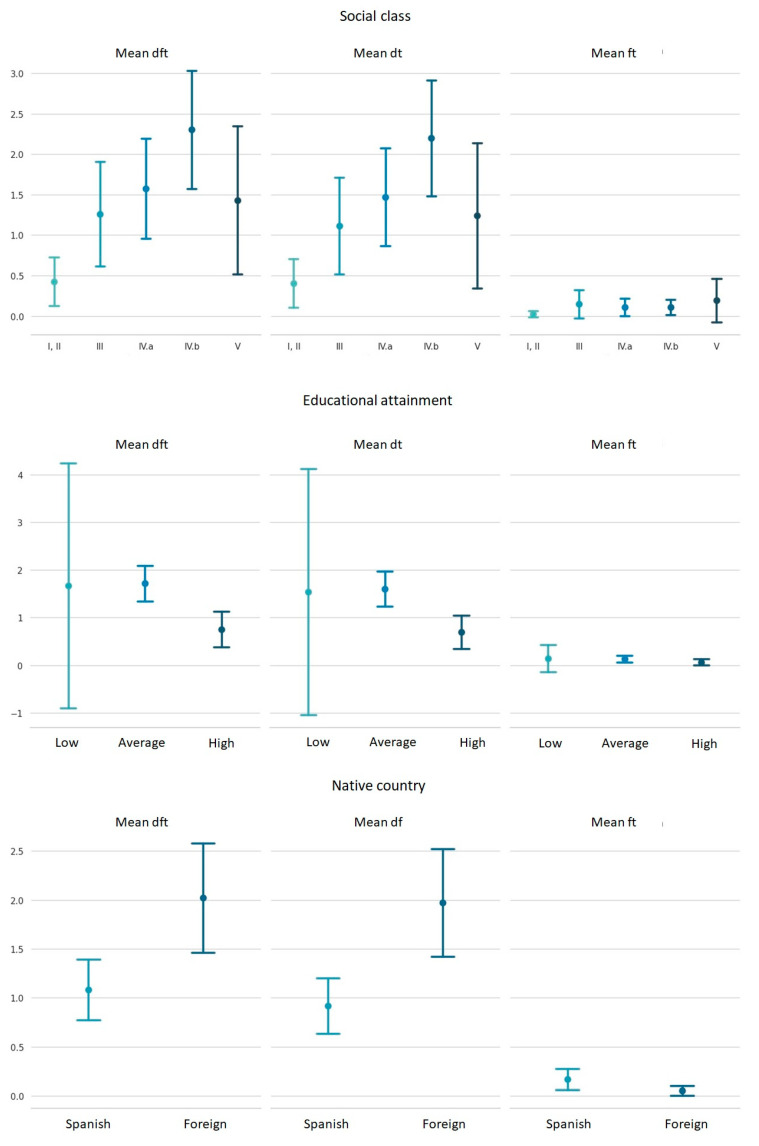
Graphical representation of the mean of the dft index, its carious component (dt) and its filled component (ft) in the total sample by social status, educational level and native country.

**Figure 5 healthcare-11-02344-f005:**
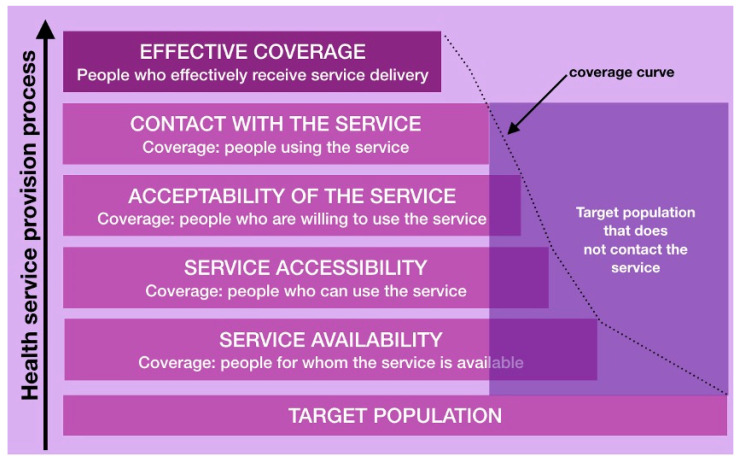
Tanahashi effective coverage [32].

**Table 1 healthcare-11-02344-t001:** Distribution of the total sample by sex and age range, taking into account the social status, educational level and country of origin of the parents.

SC ^1^	Boys n (%)	Girls n (%)	3 Years n (%)	4 Years n (%)	5 Years n (%)	Total n (%)
Total	175 (100)	168 (100)	99 (100)	122 (100)	122 (100)	343 (100)
Social status						
I	9 (5.1)	9 (5.4)	5 (5.1)	7 (5.7)	6 (4.9)	18 (5.2)
II	19 (0.9)	13 (7.7)	10 (10.1)	15 (12.3)	7 (5.7)	32 (9.3)
III	29 (16.6)	27 (16.1)	17 (17.2)	17 (13.9)	22 (18.1)	56 (16.3)
IVa	36 (20.6)	52 (31.0)	32 (32.3)	28 (23.0)	28 (23.0)	88 (25.7)
IVb	49 (28.0)	47 (28.0)	26 (26.2)	27 (22.1)	43 (35.2)	96 (28.1)
V	28 (16.0)	19 (11.3)	8 (8.1)	24 (19.7)	15 (12,3)	47 (13.7)
VI	5 (2.9)	1 (0.6)	1 (1.0)	4 (3.3)	1 (0.8)	6 (1.7)
Educational level						
Not educated	1 (0.6)	1 (0.6)	0 (0.0)	0 (0.0)	2 (1.6)	2 (0.6)
1st grade	7 (4.0)	6 (3.6)	1 (1.0)	12 (9.8)	0 (0.0)	13 (3.8)
2nd grade 1st cycle	73 (41.7)	77 (45.8)	42 (42.4I)	46 (37.8)	62 (50.8)	150 (43.7)
2nd grade 2nd cycle	53 (30.3)	54 (32.1)	36 (36.4)	32 (26.2)	39 (32.0)	107 (31.2)
3rd grade	41 (23.4)	30 (17.9)	20 (20.2)	32 (26.2)	19 (15.6)	71 (20.7)
Country of origin						
Spain	89 (50.9)	88 (52.4)	47 (47.5)	67 (54.9)	63 (51.6)	177 (51.6)
Other country UE ^2^	7 (4.0)	5 (3.0)	2 (2.0)	2 (1.7)	8 (6.6)	12 (3.5)
Other country Europe	1 (0.6)	1 (1.6)	1 (1.0)	1 (0.8)	0 (0.0)	2 (0.6)
Canada or EEUU	0 (0.0)	0 (0.0)	0 (0.0)	0 (0.0)	0 (0.0)	0 (0.0)
Other country America	49 (28.0)	49 (29.2)	39 (39.4)	32 (26.2)	27 (22.1)	98 (28.6)
Asia	0 (0.0)	3 (1.8)	0 (0.0)	1 (0.8)	2 (1.7)	3 (0.9)
Africa	26 (14.9)	21 (12.5)	9 (9.1)	16 (13.1)	22 (18.0)	47 (13.6)
Oceania	0 (0.0)	0 (0.0)	0 (0.0)	0 (0.0)	0 (0.0)	0 (0.0)
Unknown	3 (1.7)	1 (0.6)	1 (1.0)	3 (2.5)	0 (0.0)	4 (1.2)

^1^ Sociodemographic characteristics. ^2^ European country.

**Table 2 healthcare-11-02344-t002:** Prevalence, mean and 95% confidence interval (CI) of the dft index and its carious component (dt) in the total sample by age range and sex.

	dft > 0 (n) % (CI 95%)	dt > 0 (n) % (CI 95%)	dft Mean (IC-95%)	dt Mean (IC-95%)
Age				
3	25	25	0.91	0.88
(n = 99)	25.25 (16.54; 33.96)	25.25 (16.54; 33.96)	(0.48; 1.34)	(0.46; 1.30)
4	30	26	1.14	0.99
(n = 122)	24.59 (16.84; 32.34)	21.31(13.94; 28.68)	(0.64; 1.64)	(0.51; 1.48)
5	66	63	2.38	2.25
(n = 122)	54.10 (45.13; 63.07)	51.64 (42.65; 60.63)	(1.77; 2.98)	(1.65; 2.84)
*p*	<0.001	<0.001	<0.001	<0.001
Sex				
Boys	65	63	1.72	1.61
(n = 175)	37.14 (29.91; 44.37)	36.00 (28.82; 43.18)	(1.24; 2.20)	(1.14; 2.09)
Girls	56	51	1.30	1.19
(n = 168)	33.33 (26.13; 40.54)	30.36 (23.33; 37.38)	(0.91; 1.69)	(0.82; 1.57)
Total	121	114	1.51	1.41
(n = 343)	35.8 (30.19; 40.36)	33.24 (28.23; 38.25)	(1.20; 1.80)	(1.10; 1.71)
*p*	0.462	0.269	0.182	0.172

**Table 3 healthcare-11-02344-t003:** Prevalence mean ± standard deviation (SD) and 95% confidence interval of the dft index (and its dt and ft components), for the following variables: social level, educational level and parental origin.

	dft > 0 n (%)	dt > 0 n (%)	ft > 0 n (%)	dft Mean ± SD (IC-95%)	dt Mean ± SD (IC-95%)	ft Mean ± SD (IC-95%)
Social status						
I, II	10 (20.0)	9 (18.0)	1 (2.0)	0.42 ± 1.07	0.40 ± 1.07	0.02 ± 0.14
(n = 50)				(0.12; 0.72)	(0.10; 0.70)	(−0.02; 0.06)
III	17 (30.4)	16 (28.6)	3 (5.4)	1.25 ± 2.40	1.11 ± 2.24	0.14 ± 0.64
(n = 56)				(0.61; 1.90)	(0.51; 1.71)	(−0.03; 0.32)
IVa	33 (37.5)	31 (35.2)	5 (5.7)	1.57 ± 2.92	1.47 ± 2.86	0.10 ± 0.50
(n = 88)				(0.95; 2.19)	(0.86; 2.07)	(−0.00; 0.21)
IVb	45 (46.9)	44 (45.8)	6 (6.3)	2.30 ± 3.60	2.20 ± 3.53	0.10 ± 0.45
(n = 96)				(1.57; 3.03)	(1.48; 2.91)	(−0.00; 0.20)
V	14 (29.8)	12 (25.5)	3 (6.4)	1.42 ± 3.13	1.23 ± 3.05	0.19 ± 0.92
(n = 47)				(0.51; 2.34)	(0.34; 2.13)	(−0.08; 0.46)
Total	119 (35.3)	112 (33.2)	18 (5.3)	1.53 ± 2.95	1.42 ± 2.87	0.10 ± 0.56
(n = 337)				(1.22; 1.85)	(1.12; 1.73)	(−0.05; 0.17)
*p*	0.016	0.007	0.847	0.006	0.006	0.644
Educational level						
Under	4 (26.7)	3 (20.0)	1 (6.7)	1.67 ± 4.65	1.53 ± 4.67	0.13 ± 0.52
(n = 15)				(−0.91; 4.24)	(−1.05; 4.12)	(−0.15; 0.42)
Medium	99 (38.5)	94 (36.6)	14 (5.4)	1.72 ± 3.06	1.60 ± 2.99	0.12 ± 0.61
(n = 257)				(1.34; 2.09)	(1.23; 1.963)	(0.05; 0.20)
High	18 (25.4)	17 (23.9)	3 (4.2)	0.75 ± 1.59	0.70 ± 1.47	0.06 ± 0.29
(n = 71)				(0.37; 1.12)	(0.34; 1.04)	(−0.01; 0.12)
Total	121 (35.5)	114 (32.2)	18 (5.2)	1.51 ± 2.93	1.40 ± 2.85	0.11 ± 0.55
(n = 343)				(1.20; 1.82)	(1.10; 1.71)	(0.05; 0.17)
*p*	0.094	0.073	0.891	0.046	0.060	0.678
Country of origin						
Spanish	55 (31.1)	50 (28.2)	13 (7.3)	1.08 ± 2.11	0.92 ± 1.90	0.16 ± 0.71
(n = 177)				(0.77; 1.39)	(0.63; 1.20)	(0.06; 0.27)
Foreign	65 (40.1)	63 (38.9)	5 (3.1)	2.02 ± 3.59	1.97 ± 3.57	0.05 ± 0.31
(n = 162)				(1.46; 2.58)	(1.42; 2.52)	(0.05; 0.17)
Total	120 (35.4)	113 (33.3)	18 (5.3)	1.53 ± 2.94	1.42 ± 2.87	0.11 ± 0.56
(n = 339)				(1.21; 1.84)	(1.11; 1.73)	(0.05; 0.17)
*p*	0.082	0.038	0.081	0.003	0.001	0.059

**Table 4 healthcare-11-02344-t004:** Percentage and 95% confidence interval of the restoration index for the social level variable.

Social Class	RI% (IC 95%)	*p*
I, II	10.00 (0.00; 32.62)	
III	9.52 (0.00; 22.62)	0.586
IVa	7.81 (0.00; 16.54)	
IVb	6.15 (0.49; 11.80)	
V	10.05 (0.00; 41.43)	

**Table 5 healthcare-11-02344-t005:** Percentage and 95% confidence interval of the restorative index for the variable educational level.

Educational Level	RI% (IC 95%)	*p*
Under	25.00 (0.00; 104.56)	
Medium	8.21 (3.40; 13.02)	0.426
High	8.33 (0.00; 20.48)	

**Table 6 healthcare-11-02344-t006:** Percentage and 95% confidence interval of the restorative index for the variable origin of the parent.

Country of Origin	RI% (IC 95%)	*p*
Spanish	13.75 (5.50; 22.00)	0.050
Foreign	4.72 (0.03; 9.41)	

**Table 7 healthcare-11-02344-t007:** Relationship of the values 0, 1, 2, 3 and 4 of the plaque index (PI) and the gingival index (GI) with the following variables: social status, educational level and parental origin. Relationship between PI and social status (*p* = 0.043). GI and social status (*p* = 0.038).

	PI 0 n (%)	PI 1 n (%)	PI 2 n (%)	PI 3 n (%)	*p*	GI 0 n (%)	GI 1 n (%)	GI 2 n (%)	GI 3 n (%)	*p*
Social status					0.043					0.038
I, II	0 (0.0)	45 (90.0)	5 (10.0)	0 (0.0)		9 (18.0)	38 (76.0)	3 (6.0)	0 (0.0)	
III	0 (0.0)	55 (98.2)	1 (1.8)	0 (0.0)		4 (7.1)	52 (92.9)	0 (0.0)	0 (0.0)	
IVa	0 (0.0)	81 (92.0)	7 (8.0)	0 (0.0)		7 (8.0)	81 (92.0)	0 (0.0)	0 (0.0)	
IVb	1 (1.0)	84 (87.5)	10 (10.4)	1 (1.0)		10 (10.4)	84 (87.5)	2 (2.1)	0 (0.0)	
V	1 (2.1)	41 (87.2)	2 (4.3)	3 (6.4)		8 (17.0)	36 (76.6)	3 (6.4)	0 (0.0)	
Total	2 (0.6)	306 (90.8)	25 (7.4)	4 (1.2)		38 (11.3)	291 (86.4)	8 (2.4)	0 (0.0)	
Educational level					0.283					0.521
Under	0 (0.0)	13 (86.7)	1 (6.7)	1 (6.7)		2 (13.3)	12 (80.0)	1 (6.7)	0 (0.0)	
Medium	2 (0.8)	236 (91.8)	16 (6.2)	3 (1.2)		29 (11.3)	224 (87.2)	4 (1.6)	0 (0.0)	
High	0 (0.0)	63 (88.7)	8 (11.3)	0 (0.0)		7 (9.9)	61 (85.9)	3 (4.2)	0 (0.0)	
Total	2 (0.6)	312 (91.0)	25 (7.3)	4 (1.2)		38 (11.1)	297 (86.6)	8 (2.3)	0 (0.0)	
Country of origin					0.681					0.694
Spanish	1 (0.6)	158 (89.3)	16 (9.0)	2 (1.1)		21 (11.9)	151 (85.3)	5 (2.8)	0 (0.0)	
Foreign	1 (0.6)	150 (92.6)	9 (5.6)	2 (1.2)		16 (9.9)	143 (88.3)	3 (1.9)	0 (0.0)	
Total	2 (0.6)	308 (90.9)	25 (7.4)	4 (1.2)		37 (10.9)	294 (86.7)	8 (2.4)	0 (0.0)	

## Data Availability

The data presented in this study are available upon request from the corresponding author (M.D.-P.).

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
