# Peer review of "Relevance of Sociocultural Inequalities and Parents’ Origins in Relation to the Oral Health of Preschoolers in Lanzarote, Spain"

_healthcare, 2023, doi:10.3390/healthcare11162344_

Round 1

Reviewer 1 Report

In the present work, Prieto-Regueiro et al try to explain the relevance of sociocultural inequalities and parent's origin in the oral health of preschoolers in Lanzarote, Spain. There are some questions that should be explained.

1. The English grammar and writing style throughout the manuscript should be revised.

For example,

Title, ‘The Oral Health’ should been changed to ‘the Oral Health’.

Line 105, ‘229 The sample’ should be revised. Numbers are generally not at the beginning of a sentence.

Line 132, ‘According to this:’, this or these?

Line 144, ‘We defined the maternal’. In general, scientific papers are written in the third-person manner rather than the first person. Please check this throughout the manuscript.

2. Abstract section should be revised.

Knowledge of background and conclusion should be included. Please unify the writing manner, ‘dft (decayed and filled teeth), dt (decayed teeth), ft (filled teeth), restorative index (RI), plaque index (PI) and gingival index (GI)’

3. Introduction section should be revised.

Some paragraphs are too short, should be incorporated. There are 5-7 references cited for one issue, References 11-15, 11-17, 19-24, 25-31. Please revise these.

4. Etical aspects subsection

There are some sentences are repeated with ‘Institutional Review Board Statement’.

5. Result section

Line 251, ‘(<0.001)’ may be changed to (P<0.001).

Figure 2 should be revised. The words should be centered.

Lines 259, 293, 445, there are three Figure 3.

6. Conclusions

Line 466, there are so many ‘low social class (IVb)’, or ‘low social level (IVb)’?

Conclusion section should refined.

Reference section

Some references are not suitable for this Journal, references 1, 7, 20, 28…..

Some new references are related to this manuscript. For example,

Lam PPY, Chua H, Ekambaram M, Lo ECM, Yiu CKY. Does Early Childhood Caries Increase Caries Development among School Children and Adolescents? A Systematic Review and Meta-Analysis. Int J Environ Res Public Health. 2022;19(20):13459.

Hong CL, Thomson WM, Broadbent JM. Oral Health-Related Quality of Life from Young Adulthood to Mid-Life. Healthcare. 2023;11(4):515.

Moderate editing of English language required.

Author Response

Dear reviewer, we greatly appreciate the comments made regarding the content of the article.

Reviewer 2 Report

Thank you for allowing me to review this scientific manuscript whose objective was to determine the relevance of sociocultural inequalities and parental origin with respect to oral health in preschool children.

Manuscript requires revision before publication:

1. ABSTRACT - please clarify the meaning of "cod index". Please make the conclusion more specific.

2. INTRODUCTION - please standardize the document for the World Health Organization (OMS or WHO).

3. METHODOLOGY - please explain the following - "ZBS, ACEI and ACEII. 229"? The methodology is unclear and it is difficult to follow, which questionnaire, how many questions, who designed it, validated? How was the sample size calculated? Statistical data processing is inadequately presented?

4. RESULTS - due to the unclear methodology, the results cannot be monitored either

5. DISCUSSION - what is fig 3 in the discussion, it is the same for the introduction. What are the strengths and limitations of this study?

Author Response

(The authors gave the same response as above.)

Reviewer 3 Report

Please comment on the meaning of the abbreviations, it is difficult to follow the content of the text

It does not seem to me that figure 2 contributes more than what already appears in table 1. I think that the text would improve if figure 2 were transferred to an annex or eliminated

In Table 2, it is not clear to me for which parameter or parameters the confidence intervals are constructed, nor the probability model that the estimators and distributions follow. Nor do I understand why the critical level (p) has been calculated, if what is being proposed are confidence intervals. I consider that table 2 requires further explanation so that it is better understood by the reader.

Figure 3 seems to me that it provides very little information about table 2 and I think it is redundant.

There are two figures denoted as "figure 3"

Table 3, like Table 2, is not clear. Please explain it further and clarify the assumptions used to make the intervals for the mean, for example, is the variable normal? Is the variance known? What probability model does the estimator follow? These explanations are necessary to know if the estimates are correct. I think there should be a higher level of statistical explanation.

In Table 3, does it make sense that the intervals for the mean of ft have negative endpoints?

In table 3, explain what utility "p" has in this estimate

Why does table 4 have a different format from the previous ones?

Why are tables 4 and 5 presented together, since they refer to the same index?

I do not understand the content of table 6. It requires further explanation.

I would ask the authors to clarify the results of the application of the ANOVA test. Are the assumptions verified to be able to apply this test? Please, the verification of the assumptions for the correct application of the ANOVA test should be clarified.

Authors should do more to clarify the statistical assumptions on which their analysis is based, as well as the analysis itself, as it is not clear.

Author Response

(The authors gave the same response as above.)

Reviewer 4 Report

The sociodemographic characteristics of the parents of the preschoolers studied distributed by sex and age are shown in Table 1. A sentence containing an inference about the Table 1 should be added. Also a sentence containing an inference about the Tables 2-5 should be added. 

The representation of the paper is very good. The quality is OK.

Author Response

(The authors gave the same response as above.)

Round 2

Reviewer 1 Report

Thanks for author’s responses. However, in general, scientific papers are written in the third-person manner rather than the first person. Please check this throughout the manuscript.

For example,

Line 50, ‘we have a conceptual framework’.

Line 57, ‘We would find’

Line 77, ‘we should not lose’

Line 96, ‘We justify this research’

Line 117, ‘we proceeded to’

Line 145, ‘we used the abbreviated’

Line 163, ‘we based ourselves’

………….

Minor editing of English language required.

Author Response

(The authors gave the same response as above.)

Reviewer 2 Report

Figure 2 shows the same table as Table 2, while Figure 3 shows Table 3 and Figure 4 shows Table 4. This is a repeat of the results.

What is the purpose of figure 5 in the discussion where the results are explained and commented on. The table is taken from another document and is irrelevant.

Author Response

(The authors gave the same response as above.)

Reviewer 3 Report

I think this paper can be published in its present form. All suggestions have been attended

Author Response

(The authors gave the same response as above.)
